# Electrochemical direct air capture of $CO_2$ using neutral red as reversible redox-active material

Hyowon Seo [1] & T. Alan Hatton [1] ✉

Direct air capture of carbon dioxide is a viable option for the mitigation of $CO_2$ emissions and their impact on global climate change. Conventional processes for carbon capture from ambient air require 230 to 800 kJ thermal per mole of $CO_2$, which accounts for most of the total cost of capture. Here, we demonstrate electrochemical direct air capture using neutral red as a redox-active material in an aqueous solution enabled by the inclusion of nicotinamide as a hydrotropic solubilizing agent. The electrochemical system demonstrates a high electron utilization of 0.71 in a continuous flow cell with an estimated minimum work of 35 $kJ_e$ per mole of $CO_2$ from 15% $CO_2$. Further exploration using ambient air (410 ppm $CO_2$ in the presence of 20% oxygen) as a feed gas shows electron utilization of 0.38 in a continuous flow cell to provide an estimated minimum work of 65 $kJ_e$ per mole of $CO_2$.

Increased atmospheric carbon dioxide ($CO_2$) concentration owing to the burning of fossil fuels is considered the major factor in recent global warming and climate change[1]. The atmospheric $CO_2$ concentration has increased continuously from the preindustrial level of 280 ppm to 410 ppm in 2021, induced by human activities which currently emit close to 40 billion tons of $CO_2$ annually[2]. Carbon capture and storage (CCS) technologies have been recognized as one of the important strategies to slow down changes in global climate patterns by effectively lowering $CO_2$ discharges[3]. Conventional $CO_2$ capture has addressed $CO_2$ emissions from large point sources, such as fossil-fuel power stations and chemical plants. Since a considerable portion of $CO_2$ emissions is derived from mobile sources (e.g., 29% from transportation), direct air capture (DAC), in which $CO_2$ is captured directly from ambient air, is considered to be an important and viable option for reducing atmospheric $CO_2$ levels[4]. While there are advantages of DAC in its potential to address emissions from distributed sources, the development and application of DAC processes have been restricted by their high operation cost. Currently, the energy requirements of the leading DAC technologies are ~500 to 800 kJ thermal per mole of $CO_2$ using sodium hydroxide scrubbing/lime causticization systems[5] and ~238 to 317 kJ thermal per mole of $CO_2$ using amine-functionalized solid sorbents[6]. Recent analyses suggest that the energy requirement target should be less than 400 kJ thermal per

mole of $CO_2$ (equivalent to 120 kJ/mol of electrical energy with Carnot efficiency of 0.3) by $CO_2$-neutral power sources to be viable in order to be $CO_2$ negative[5].

To overcome these limitations of high energy requirements encountered by the systems using thermal energy, electrochemical systems have come to be recognized as a feasible option due to their potentially lower energy consumption under milder conditions of room temperature and pressure, as highlighted in recent review articles (Supplementary Table 1)[7,8]. Such electrochemical systems are based on pH swing[8–11] and reversible redox-active capturing agents[12–21]. Most systems involving redox-active capture agents suffer from oxygen sensitivity which hampers their larger-scale application[9,22]. In this regard, our group has been searching for oxygen-insensitive redox-active compounds that can capture $CO_2$ under highly dilute conditions (i.e., ambient air).

Phenazines and phenothiazines are electron-rich heterocyclic organic π-systems that often favorably fulfill the requirements of organic redox-active compounds with their reversibility in aqueous systems. In this respect, there has been extensive research employing phenazine and phenothiazine derivatives in redox-flow batteries[23] and electrochemical carbon capture[9,10,24] via proton-coupled electron transfer (PCET) in aqueous systems. Although several efforts to engage these compounds in electrochemical direct air capture with potentially low energy consumption have been reported, these systems suffered

[1]Department of Chemical Engineering, Massachusetts Institute of Technology, Cambridge, MA 02139, USA. ✉e-mail: tahatton@mit.edu

from oxygen sensitivity and required synthetic modification to overcome low aqueous solubility[9,10]. For these reasons, none of the phenazine- and phenothiazine-based systems was reported to demonstrate capture of $CO_2$ from ambient air, to the best of our knowledge. Direct air capture using a reversible electrochemical system with low energy consumption benefiting from the reversible PCET redox couples in an aqueous solution can be achieved by eliminating oxygen sensitivity of the redox-active compound and enhancing its aqueous solubility. Here, we report electrochemical direct air capture of $CO_2$ using neutral red (NR), a commercial organic dye molecule, as an oxygen-insensitive organic redox-active compound (Fig. 1a) in the presence of nicotinamide (NA) as a hydrotropic agent to increase its solubility in the aqueous system (Fig. 1b)[25]. Other commercial phenothiazine compounds such as toluidine blue (TB), and thionin (TN) could not satisfy the need for oxygen insensitivity (Fig. 1a), and therefore were not considered further for this task. The minimum energy requirements are estimated to be 35 kJ$_e$/mol of $CO_2$ with a 15% $CO_2$ feed, and 65 kJ$_e$/mol for direct air capture.

The proposed working scheme for electrochemical carbon capture by the NR/leuco-neutral red (NRH2) redox system is illustrated in Fig. 1c. NRH2, the reduced product from NR, was formed on application of a suitable electrochemical potential with basification of the aqueous solution to pH 12 (experimentally obtained). Then the $CO_2$-rich gas stream was introduced to saturate and consequently acidify the solution. Subsequent electrochemical oxidation of the $CO_2$-saturated solution regenerated NR and released free $CO_2$. The conjugate base NR would mainly participate in the redox process because the system operates in the pH range of 6–12. The amount of hydroxide ion formed by electrochemical reduction of the NR/NRH equilibrium (pKa = 6.8)[26] mixture can be estimated to be 1.2 to 1.67 equivalent depending on the pH values of the starting solutions (Supplementary equations 1–4).

Although the aqueous solubilities of phenazines and phenothiazines would be low in electrolyte solutions, the inclusion of 1 M of NA as a hydrotropic agent, which is often used to increase the water solubility of pharmaceuticals[25], can increase the solubility of NR in 0.5 M potassium chloride (KCl) aqueous solution from 46 mM to 306 mM (Fig. 1b). Cyclic voltammetry (CV) curves recorded to compare the redox activity of NR/NRH2 in 1 M NA solution to those in the absence of NA under 1 atm $N_2$ and 1 atm $CO_2$ atmospheres are shown in Fig. 1c, where the first set of CV curves obtained in the absence of NA is displayed in blue curves. Under the $N_2$ atmosphere, NR showed a cathodic peak at −0.78 V vs Ag/AgCl and an anodic peak at −0.68 V vs Ag/AgCl (pastel blue curve). When the electrolyte solution was saturated with $CO_2$, the voltammogram showed a positively shifted single cathodic peak at −0.65 V and an anodic peak at −0.54 V (blue curve). The second set of CV curves, shown in red, were recorded in the presence of NA. Under the $N_2$ atmosphere, two sets of cathodic and anodic peaks appeared at −0.60 V and −0.50 V and at −0.97 V and −0.82 V, respectively (pastel red curve). These are attributed to the stepwise two single-electron transfers[27]. When the $CO_2$ was introduced to the electrolyte solution with adjustment of the pH value to pH 7, the two cathodic peaks merged to show a single peak at −0.78 V with slight shifts in the two anodic peaks to −0.61 V and −0.52 V, respectively (red curve). CV curves of NR show quasi-reversibility in the redox activity of NR/NRH2 in the presence and absence of NA. The electrochemical carbon capture system in this manuscript was demonstrated using a 50 mM NR solution in the presence of 1 M NA as a hydrotropic agent.

In this manuscript, we describe continuous flow electrochemical $CO_2$ capture with the NR/NRH2 redox system (Fig. 1d). NR was reduced electrochemically to provide NRH2 with an increase in the pH of the solution (path a). The basic aqueous solution was pumped to the reservoir where an air or $CO_2$-rich gas stream was introduced (path b). The $CO_2$-saturated solution was then pumped to the anodic chamber where electrochemical oxidation led to regeneration of NR and release

of free $CO_2$ (path c). Then the resulting solution was transferred to the anolyte reservoir to discharge $CO_2$ and close the flow cycle (path d). The continuous operation of the flow cell was demonstrated with 15% $CO_2$ and ambient air.

## Results

### Cyclic voltammetry of NR

The electrochemistry of the NR aqueous solution was examined by CV at various pH levels between 6 and 12 (Fig. 2a). Two sets of peaks were observed, corresponding to the first and second single-electron transfers to NR via an H e e H mechanism[27]. The first single-electron reduction peak appears at −0.61 V with the second reduction peak at −0.94 V and two single-oxidation peaks at −0.68 V and −0.52 V vs Ag/AgCl at pH 6. As the solution pH increases, the first single-electron reduction peak decreases and merges with the second electron transfer peak, which is consistent with the H e e H mechanism. The peak currents ($i_{pc}$) in the CVs shown in Fig. 2b for the first and second single-electron reductions vary linearly with the square root of the scan rate (Fig. 2c) at pH 7, the differences in slope indicating that the first reduction is kinetically slower than the second reduction, which also supports the H e e H mechanism at pH>pKa. Based on the experimental pH measurement at each step and results from the CV experiment in Figs. 1c, 2a the minimum potential gap for the electrochemical swing process should be 0.42 V for the first cycle without $CO_2$ around during reduction. The minimum potential gap of 0.26 V in the cyclic system in which $CO_2$ is dissolved in the solution is smaller based on the CV in Fig. 1c. The merging of the first and second reduction peaks in the presence of $CO_2$ suggests that pre-association of $CO_2$ and NR would facilitate electron transfer. The NR/NRH2 redox reaction demonstrates good reversibility and excellent redox durability, with no significant decay in the peak current after 100 CV cycles (Fig. 2d).

### Electrochemical release of $CO_2$ using the NR/NRH2 redox system

A bench-scale setup using an electrochemical H-cell was constructed for $CO_2$ capture and release in the NR/NRH2 redox system (Fig. 3a). The system was equipped with an anion exchange membrane separating two 5 mL reaction chambers, carbon felt as a working electrode, and a stainless-steel wire electrode for an arbitrary reaction in the counter chamber. The 4 mL reaction mixture containing 50 mM NR (200 mmol) in water in the presence of 1 M NA as a hydrotropic agent and 0.5 M LiClO4 as a supporting electrolyte was electrochemically reduced in a constant current mode at 50 mA for 695 s (equivalent to 360 mmol of electrons transferred) to yield 90% reduction of the NR to NRH2. Then a 15% $CO_2$ gas stream was introduced for 10 min to saturate the solution. The output gas flow from the $CO_2$-saturated solution upon anodic oxidation was quantified and qualified by a $CO_2$ flow meter and an FT-IR $CO_2$ sensor, respectively. Plots of the amount of $CO_2$ released by electrochemical oxidation versus electric charge are displayed in Fig. 3b. The electron utilization for the $CO_2$ release by oxidation represents the ratio between the moles of $CO_2$ released per mole of electrons transferred. $CO_2$ release at an electron utilization of 0.50 was obtained, accounting for the delayed release of $CO_2$ from the cell. Combining the voltage difference (0.42 V) between the peak potentials from the CV measurements with the electrochemical electron utilization of 0.50 during $CO_2$ release, we estimated that the minimum energy requirement of 81 kJ$_e$/mol in batch (Supplementary equations 5 and 6).

### $CO_2$ absorption dynamics of NRH2 solutions

The dynamics of $CO_2$ absorption by 1 mL of a 50 mM NRH2 solution depended on the $CO_2$ concentration, as shown in Fig. 4a for 1, 4, and 15% $CO_2$ feeds. A saturation absorption of 78 μmol of $CO_2$ in equilibrium with 15% $CO_2$ was observed, corresponding to 1.56 equivalents

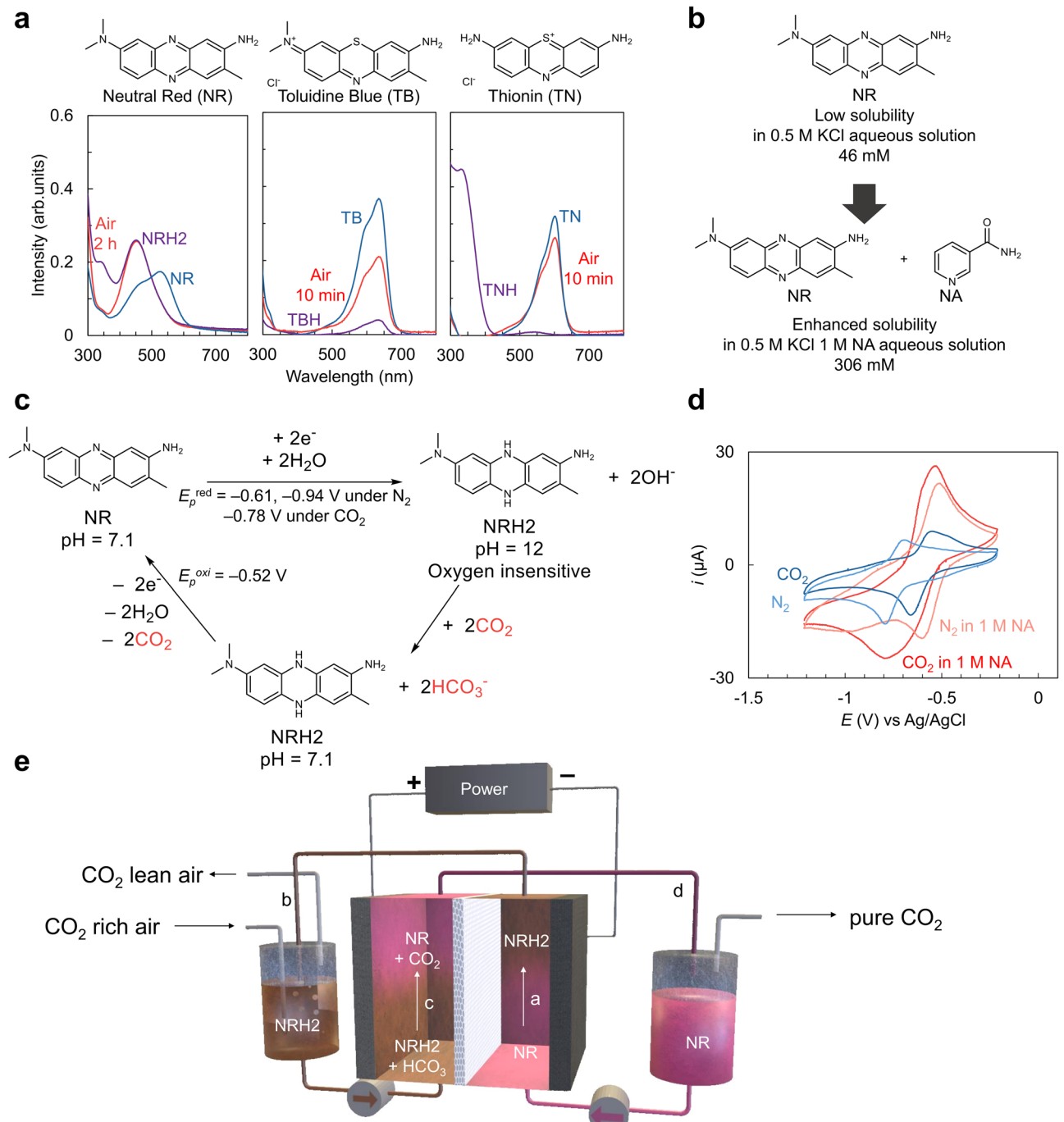

**Fig. 1 | Electrochemical carbon capture using a redox pair of NR/NRH2 in an aqueous solution. a** UV–vis spectra for neutral red (NR), toluidine blue (TB), and thionin (TN) in an aqueous solution during tests for air sensitivity. Solutions containing the reduced organic dye compounds (50 mM, 1 mL) were bubbled with air for 10 min (TBH, TNH) or 2 h (NRH2) at a flow rate of 3 mL/min. NRH2 leuco-neutral red, TBH leuco-toluidine blue, TNH leuco-thionin. **b** NR solubility enhancement in water with the inclusion of 1 M nicotinamide (NA) as a hydrotropic agent. KCl potassium chloride. **c** Scheme of the reversible electrochemical $CO_2$ capture and release using the NR/NRH2 redox system in water. Potentials are versus Ag/AgCl.

**d** The cyclic voltammograms of 5 mM NR under nitrogen (pastel blue curve, pH 6) and $CO_2$ (blue curve, pH 7) in water with 0.1 M lithium perchlorate ($LiClO_4$) as a supporting electrolyte, and those of 10 mM NR under nitrogen (pastel red curve, pH 6) and $CO_2$ (red curve, pH 7) in water with 1 M NA as a hydrotropic agent and 0.1 M $LiClO_4$ as a supporting electrolyte. All CV curves were recorded at room temperature at a scan rate of 50 mV/s with a glassy carbon working electrode. Potentials were recorded versus Ag/AgCl as a reference electrode. **e** Overview of the continuous flow electrochemical cell with the NR/NRH2 redox cycle for $CO_2$ capture and release experiments.

to NRH2, consistent with the amount of hydroxide anion estimated (1.67 equivalent to NRH2, Supplementary equations 1–4). The $CO_2$ absorption levels were lower with 4 and 1% $CO_2$ gas streams, at 55 μmol (1.1 equivalents to NRH2) and 45 μmol (0.9 equivalents to NRH2), respectively. When normalized by the inlet $CO_2$ concentration, the curves superimpose in the initial absorption period, as shown in

Fig. 4b, consistent with mass transport limitations during the early stages of absorption. However, it shows lower utilization of the capacity of the solution when using lower $CO_2$ concentrations of the inlet gas streams, as would be anticipated based on mass action kinetics. Absorption was also monitored by in situ pH measurement in 5 mL of 50 mM NRH2 solution sparged with 1, 4, and 15% $CO_2$ stream at a flow

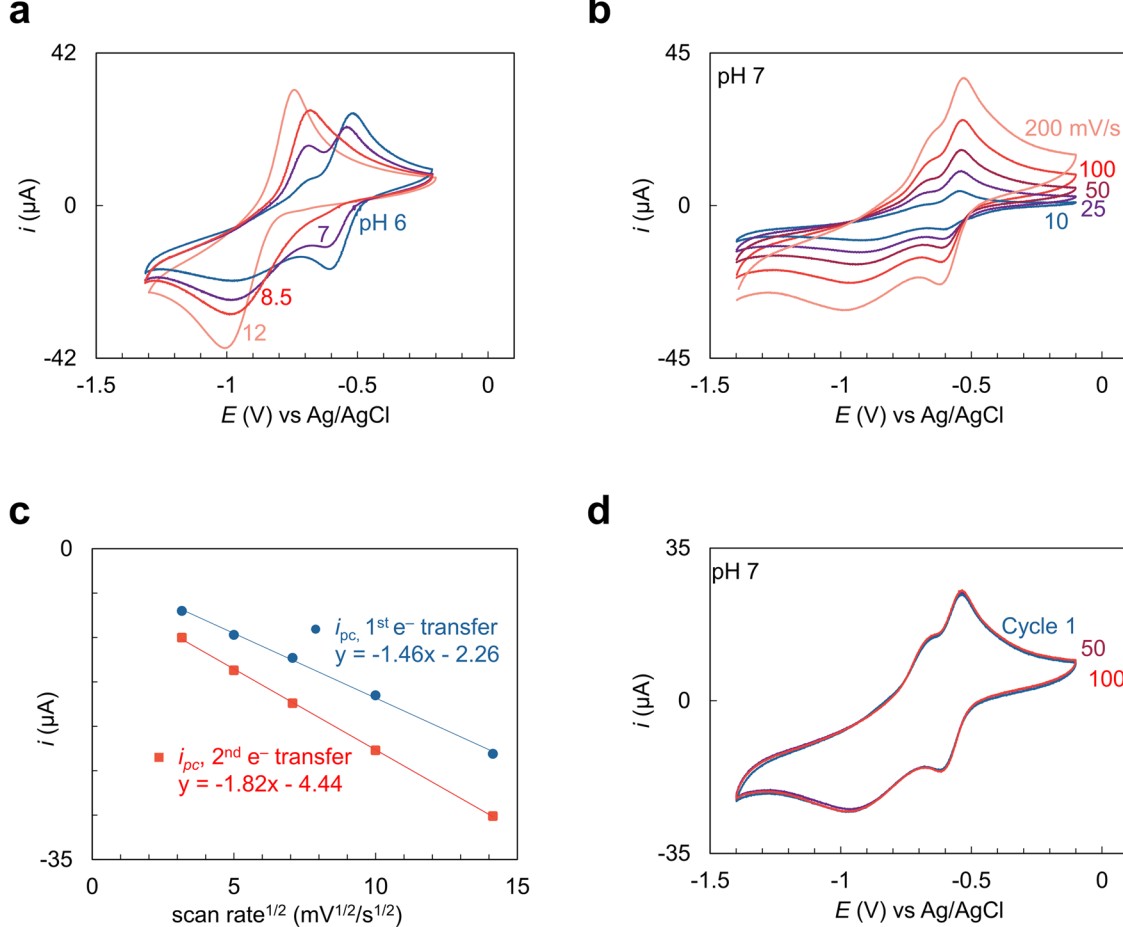

**Fig. 2 | Cyclic voltammetry of NR in aqueous solutions. a** The cyclic voltammograms of 10 mM NR at pH 6 (blue), 7 (purple), 8.5 (red), and 12 (pastel red) in 1 M NA and 0.1 M $LiClO_4$ solutions under nitrogen with a glassy carbon working electrode, at a scan rate of 50 mV/s. Potentials were recorded versus Ag/AgCl as a reference electrode. **b** Cyclic voltammograms of NR with scan rates of 10 (blue), 25 (purple), 50 (dark red), 100 (red), and 200 mV/s (pastel red). **c** Analysis of NR redox reaction of peak current ($i_{pc}$) versus the square root of scan rate ($v^{1/2}$) for the first (blue) and second reduction peaks (red). **d** 100 cyclic voltammograms of NR. Cycle 1: blue, cycle 50: purple, cycle 100: red.

rate of ca. 100 mL/min (Fig. 4c). The final pH value after saturation was 7.1 with 15% $CO_2$, 7.4 with 4% $CO_2$, and 8.1 with 1% $CO_2$ which are consistent with the results in Fig. 4a, b. We also compared the NR $CO_2$ absorption profiles with those of 50 mM solutions of amines ethylenediamine (EDA) and monoethanolamine (MEA), which are traditionally used for $CO_2$ capture (Fig. 4d–f); these electrochemically inactive amines are frequently used in comparative studies of $CO_2$ absorption[28,29]. The initial $CO_2$ absorption rate in the NRH2 solution was comparable to that in EDA, but the amount of absorbed $CO_2$ was 56% higher than that with EDA when using a 15% $CO_2$ inlet gas stream (Fig. 4d). The absorption rate by the NRH2 solution with 4% $CO_2$ was also as fast as that by EDA. However, the NRH2 solution shows slower absorption than EDA with the 1% $CO_2$ gas feed. In all cases, $CO_2$ absorption by NRH2 was faster than by the MEA solution.

**Direct air capture and stability test of NRH2 solution**
Next, we investigated the potential for electrochemical direct air capture using the NR/NRH2 redox system (Fig. 5). A 45 mM NRH2 solution prepared by electrochemical reduction was contacted with non-pretreated air for 3 h at a flow rate of ca. 120 mL/min. Electrochemical oxidation of the air-contacted solution was carried out to evaluate the direct air capture efficiency (Fig. 5a). The system presented an electron utilization during $CO_2$ release of up to 0.33, with an average value of 0.21 under the current batch conditions. On the basis of the electron utilization (0.33) during $CO_2$ release and the potential difference

obtained from the CV, the minimum work for direct air capture under the current conditions was estimated to be 123 kJe/mol (Supplementary equations 5 and 7). Although further engineering optimization is warranted, the estimated minimum energy is promising as it is in the vicinity of 400 kJ/mol thermal (equivalent to 120 kJe/mol with a Carnot efficiency of 0.3), which is considered to be the target for DAC technologies[5]. The absorption of $CO_2$ during bubbling of air was monitored by in situ measurement of pH (Fig. 5b), which dropped from 12 to 9.1 in 3 h. The initial absorption rate from the ambient air was comparable to that for a 1% $CO_2$ concentration inlet based on the pH value measurements as depicted in a normalized plot (See Supplementary Fig. 21).

The NRH2 solution was shown to be insensitive to oxygen as observed in a set of UV–vis absorption spectroscopy experiments under various conditions (Fig. 5c–f). The 50 mM NRH2 solutions prepared by electrochemical reduction in the presence of 1 M NA and 0.5 M KCl in water were bubbled with pure oxygen, nitrogen, $CO_2$, and ambient air for 24 h. The freshly prepared NRH2 solution showed two absorption peaks in the range of 300–800 nm by UV–vis absorption spectroscopy. The larger peak at 455 nm is attributed to the NRH2 and the smaller peak at 345 nm is presumably from the radical species[27]. In Fig. 5c, the peak intensity at 455 nm was maintained for 2 h when contacted with pure oxygen. Longer studies for a day and a week provided peak intensities of NRH2 lowered by 42% and 52%, respectively, in the solution sealed under an oxygen atmosphere after 24 h of

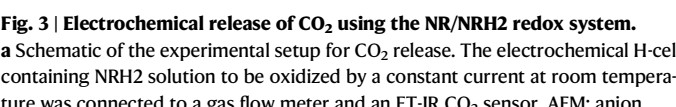

**Fig. 3 | Electrochemical release of CO₂ using the NR/NRH2 redox system.**
**a** Schematic of the experimental setup for CO₂ release. The electrochemical H-cell containing NRH2 solution to be oxidized by a constant current at room temperature was connected to a gas flow meter and an FT-IR CO₂ sensor. AEM: anion exchange membrane. **b** Solution of 4 mL of 50 mM NRH2 was oxidized at a constant current of 50 mA. The amount of released CO₂ (blue curve) and electron utilization (red curve) are shown versus the electric charge.

bubbling with oxygen. The peak at 345 nm disappeared rapidly over 30 min on contacting oxygen. As a control experiment, we introduced nitrogen to the 50 mM NRH2 solution at the same flow rate (Fig. 5d). Initial degradation of the peak intensity at 455 nm and 345 nm was slower during the first 2 h. The peak intensity measured after 24 h and a week provided the peak intensity degradation by 33% and 53%, respectively. These results suggest that the oxygen with which the solution had been in contact did not contribute significantly to the degradation of the UV–vis peak intensity of NRH2 at 455 nm, while the 345 nm peak disappeared rapidly in reaction with oxygen. We investigated possible factors that might contribute to the 455 nm peak degradation, including decomposition, reoxidation, precipitation, and polymerization. NMR studies of the NRH2 solutions indicated that the NRH2 precipitation due to its limited solubility under the current conditions led UV–vis peak degradation (See Supplementary Fig. 22). In addition to peaks from NRH2 and NA, minor peaks potentially derived from NA were observed under O₂ and N₂. Although further studies are warranted on the byproduct formation pathways, we currently believe that precipitation is one of the causes of the reduced electron utilization, albeit a minor one. We believe that the solubility can be improved by the inclusion of a higher concentration of NA or a choice of supporting electrolyte. We did not observe any major decomposition, reoxidation, or polymerization products of NRH2 by NMR studies.

A third set of UV–vis absorption measurements (Fig. 5e), this time on solutions contacted with CO₂, was carried out. Interestingly, no significant degradation of peak intensity at 455 nm was observed by UV–vis absorption spectroscopy after a day. These results indicated that the homogeneity of the NRH2 solution is better maintained under the CO₂ atmosphere possibly due to the neutral pH of the solution. The peak at 345 nm, however, disappeared rapidly in 30 min accompanied by bumps at ~490 nm and ~550 nm, which may be due to the regeneration of NR and NRH, respectively (See Supplementary Fig. 17). Based on this result, at this point CO₂ reduction by the radical species formed by electrochemical reduction under the imposed current conditions cannot be ruled out. The final set of UV–vis absorption experiments in Fig. 5f was carried out using solutions bubbled with ambient air. Although the intensities of the 455 nm peak were the same for the samples after 30 min and 2 h contact, respectively, similar levels of peak intensity degradation as observed with solutions contacted with oxygen and nitrogen were observed with longer sample storage times.

## Continuous flow operation of NR/NRH2 redox system

We constructed a continuous flow cell to process 50 mM NR solution in the presence of 1 M NA and 1 M KCl as a supporting electrolyte due to its better conductivity than that of LiClO₄ (28.3 mS/cm in 1 M LiClO₄ and 1 M NA in water vs 43.7 mS in 1 M KCl and 1 M NA in water) and higher solubility of NR in water (50 mM in 0.5 M LiClO₄ and 1 M NA vs 306 mM in 0.5 M KCl and 1 M NA) to avoid potential clogging of the tubing in the flow system. A schematic of the flow cell is shown in Fig. 6a. We developed the flow cell structure with carbon-felt electrodes in both chambers and an anion exchange membrane that divides the cell into the cathodic and anodic chambers and enables the pH difference between them to be maintained. The system is equipped with two reservoirs, one for catholyte that contains NRH2, which absorbs CO₂ from the CO₂-rich gas stream, and one for the anolyte that contains NR from which separated CO₂ would be discharged to be measured. The results of carbon capture from 15% CO₂ are displayed in Fig. 6b, c. We utilized 90% of the capacity of the system to minimize undesired side reactions under the constant current mode of operation at 50 mA with 15% CO₂. The liquid flow rate was 0.349 mL/min, giving 6.3 min of residence time in the 2.2 mL volume of each chamber. The gas output and CO₂ fraction were recorded simultaneously during the operation to show the reproducibility of CO₂ capture and release over a 12 h period, which corresponds to over eight circulations of the solution through the system. The linearity of the cumulative CO₂ released over time depicted in Fig. 6b indicates steady state operation with a constant rate of CO₂ capture and release, yielding a separation of ~350 mL of CO₂ in 12 h. In Fig. 6c, electron utilization, calculated from the CO₂ flow rate and the electric current, showed 0.41 for the first cycle and jumped to 0.71 in the second cycle with better stability. The electron utilization dropped gradually to 0.5 after 12 h. The minimum energy requirement can be estimated from the voltage gap obtained from CV experiments for the cyclic system (0.26 V) combined with the electron utility (0.71) to provide 35 kJₑ/mol of CO₂ (Supplementary equations 8 and 9).

To explore the possibility of using the NR/NRH2 flow cell for direct air capture, we performed CO₂ capture and release using ambient air as a feed gas (Fig. 6d, e). The scheme of the setup is the same as in Fig. 6a with a constant current mode of operation at 30 mA, a liquid flow rate of 0.218 mL/min (residence time of 10.1 min), and a flow rate of bubbling air of ca. 1000 mL/min. We used in-house supply air without any pretreatment before the experiments. We have included additional water injection at 1.5 mL/h to the catholyte to compensate for water evaporation by the rapid bubbling of the air through

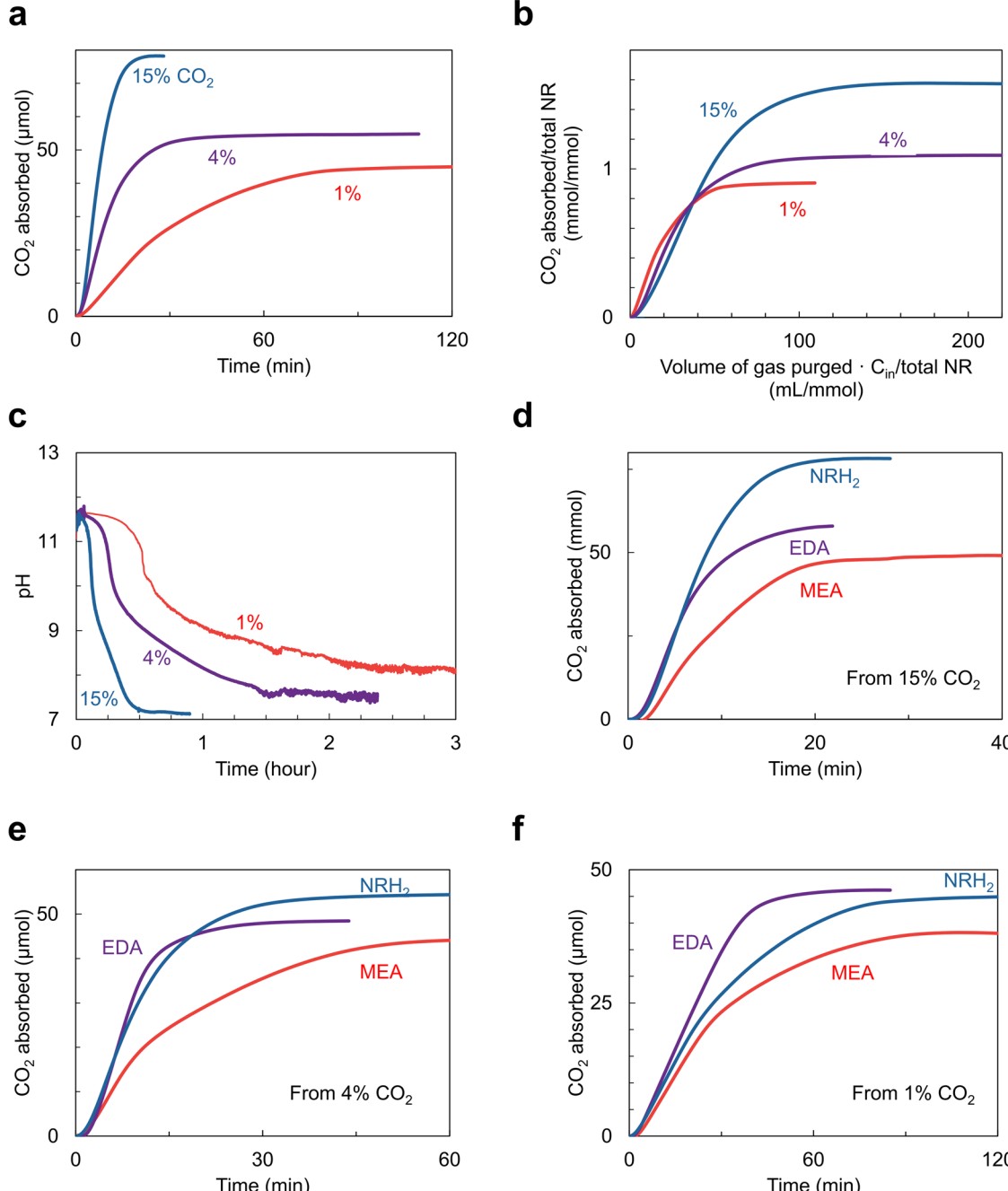

**Fig. 4 | Dynamics of CO₂ absorption by 50 mM NRH2 solutions. a** CO₂ absorption profiles at CO₂ inlet gas stream concentrations of 1 (red curve), 4 (purple curve), and 15% (blue curve). An aqueous 50 mM NRH2 solution was contacted with the gas at a flow rate of 3.3 mL/min at room temperature. **b** Normalized CO₂ absorption profiles. **c** In situ pH measurement during CO₂ bubbling. Plots of pH versus time are displayed for 5 mL of 50 mM NRH2 solution with CO₂ inlet gas stream concentrations of 1 (red curve), 4 (purple curve), and 15% (blue curve) at a flow rate of 100 mL/ min. **d** Comparison of the CO₂ absorption profile with 15% CO₂ for NRH2 (50 mM in 1 mL of water) to those for ethylenediamine (EDA, 50 mM in 1 mL of water) and monoethanolamine (MEA, 50 mM in 1 mL of water). A flow of 15% CO₂ concentration balanced by nitrogen was used at a flow rate of 3.3 mL/min. **e** Comparison of the CO₂ absorption profiles with 4% CO₂. **f** Comparison of the CO₂ absorption profiles with 1% CO₂.

the solution. Figure 6d presents the amount of released CO₂ over time. The system separated ~880 mL of CO₂ from ambient air over 96 h. As depicted in Fig. 6e, the electron utilization was increased from 0.24 to 0.32 within the first circulation time (4.8 h). The electron utilization was maintained between 0.3–0.4 for 96 h of operation. The reduction in electron utilization after ca. 43 h was due to partial clogging of the needle used to introduce air to the catholyte solution; the high air flow rate (ca. 1000 mL/min) quickly dried water at the tip of the needle, leading to precipitation of solution components, most likely NR, NA,

and KCl. Indeed, we repeatedly observed a reduction in electron utilization every 24–48 h due to this air needle clogging. On replacement of the air needle, we observed that the electron utilization increased again to the range of 0.3–0.4. The minimum energy requirement for direct air capture can be estimated from the voltage gap obtained from CV experiments for the cyclic system (0.26 V) combined with the best electron utility (0.38) to provide 65 kJₑ/mol of CO₂ (Supplementary equations 8 and 10). After the operation of direct air capture in a continuous flow, the anolyte and catholyte solutions were analyzed by

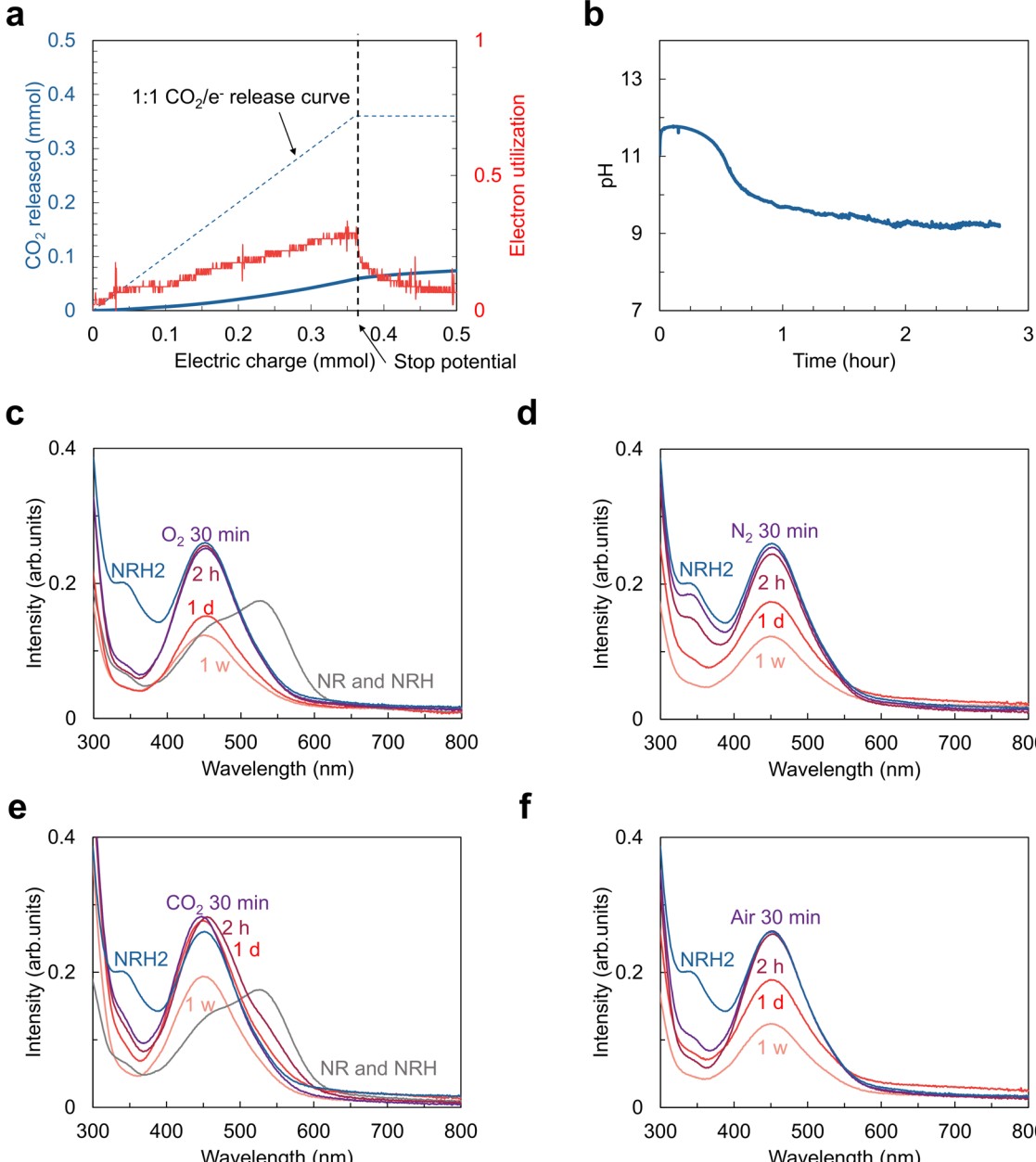

**Fig. 5 | Direct air capture and stability test of NRH2 solutions. a** $CO_2$ released by electrochemical oxidation on the application of a constant current of 50 mA to the NRH2 solution (50 mM, 4 mL) bubbled with air for 3 h at a flow rate of ca. 120 mL/min. The amount of released $CO_2$ (blue curve) and electron utilization (red curve) are shown versus electric charge. **b** In situ pH measurement versus time for 5 mL of 50 mM NRH2 solution bubbled with ambient air. **c** UV–vis spectra for the oxygen sensitivity test. The NRH2 solutions (blue curve, 50 mM, 1 mL) were bubbled with pure oxygen for 30 min (purple), 2 h (dark red), and 1 day (red) at a flow rate of 3 mL/min and sealed under oxygen for 1 week (pastel red). Gray curve represents UV–vis spectra of NR and NRH. **d** UV–vis spectra for the control experiment under nitrogen. The NRH2 solutions (blue curve, 50 mM, 1 mL) were bubbled with nitrogen for 30 min (purple), 2 h (dark red), and 1 day (red) at a flow rate of 3 mL/min and sealed under nitrogen for 1 week (pastel red). **e** UV–vis spectra for the stability test under $CO_2$. The NRH2 solutions (blue curve, 50 mM, 1 mL) were bubbled with pure $CO_2$ for 30 min (purple), 2 h (dark red), and 1 day (red) at a flow rate of 3 mL/min and sealed under $CO_2$ for 1 week (pastel red). Gray curve represents UV–vis spectra of NR and NRH. **f** UV–vis spectra for the stability test under air. The NRH2 solutions (blue curve, 50 mM, 1 mL) were bubbled with air for 30 min (purple), 2 h (dark red), and 1 day (red) at a flow rate of 5 mL/min and sealed under air for 1 week (pastel red).

[1]H-NMR to confirm no participation of NA in this redox process (See Supplementary Fig. 4).

Although further optimization and engineering efforts are required for scale-up, the NR/NRH2 redox couple meets several essential criteria for implementation in carbon capture processes, including redox stability, reasonable solubility in water, fast kinetics, and oxygen insensitivity. In addition, to operate this system with the minimum potential required on an industrial scale, it will be necessary to increase the current density and reduce the overpotential by optimizing the design of the cell structure and material.

## Discussion

In this work, we demonstrated electrochemical $CO_2$ capture and release from 15% $CO_2$ and ambient air using the NR/NRH2 redox system in an aqueous electrolyte. The estimated minimum electrochemical energy requirements in continuous flow are in the range of 35 kJe/mol

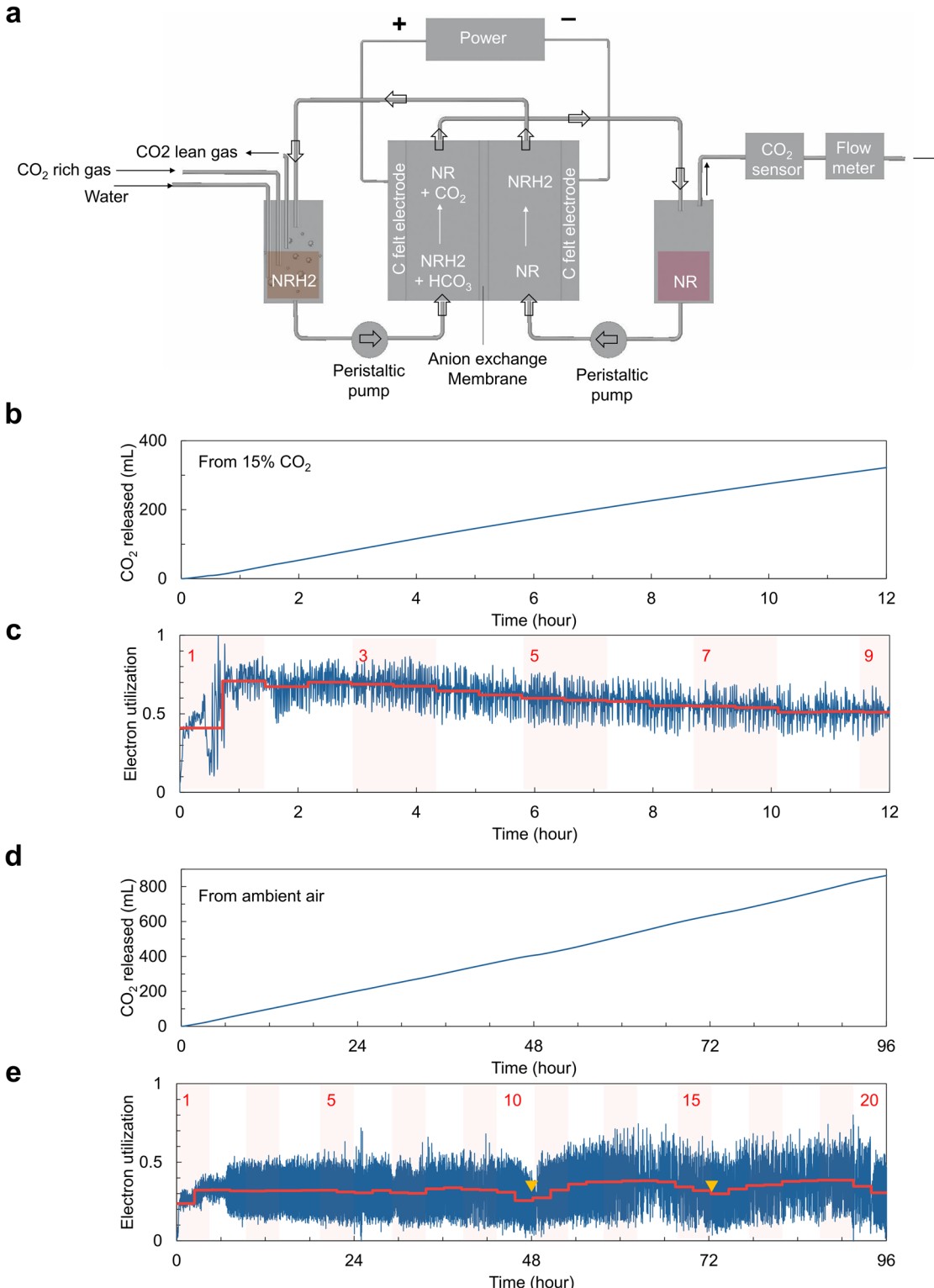

**Fig. 6 | Continuous flow operation of the NR/NRH2 redox system. a** Scheme of continuous flow electrochemical cell using the NR/NRH2 redox cycle for $CO_2$ capture and release experiments. **b** Released $CO_2$ amount over time with a feed gas composition of 15% $CO_2$. **c** Electron utilization over time with a feed gas concentration of 15% $CO_2$. Electrolytes comprised 15 mL of 50 mM NRH2 in 1 M NA and 1 M KCl and 15 mL of 50 mM NRH in 1 M NA and 1 M KCl in water. The red curve indicates the average value of electron utilization for each one-way travel (circulation time/2). The numbers in red indicate the time/circulation time. **d** Released $CO_2$ amount over time using ambient air as the feed gas. **e** Electron utilization over time with an ambient air feed. Electrolytes comprised 30 mL of 50 mM NRH2 in 1 M NA and 1 M KCl and 30 mL of 50 mM NR in 1 M NA and 1 M KCl in water. The red curve indicates the average value of electron utilization for each one-way travel (circulation time/2). The numbers in red indicate the time/circulation time. The yellow triangle mark indicates where the air needle was replaced.

of $CO_2$ using 15% $CO_2$ and 65 kJ$_e$/mol of $CO_2$ using air. The NRH2 solution exhibits a fast $CO_2$ absorption rate from 1 to 15% $CO_2$, while a higher capacity with 15% $CO_2$ was observed. Notable stability of the NRH2 solution under $O_2$ was demonstrated over 1 week. We demonstrated a robust, stable, reversible, and scalable continuous flow electrochemical cell operation with homogeneous aqueous solutions over 96 h under air. We were able to ascertain that the absorption unit can be improved for system stability in the future. Due to the slow absorption of $CO_2$ from the air (410 ppm), a high flow rate of air was necessary to saturate the catholyte solution. In our lab scale demonstration, we used ca. 1000 mL/min of airflow for the operation at 30 mA current and 20−30 mL of catholyte solution. Due to the rapid bubbling of the solution, splashing of the catholyte and evaporation of water may have caused degradation in the lifetime of the cell. We believe that a better $CO_2$ absorption unit design can improve $CO_2$ absorption rate as well as the electrochemical cell's operation time in future studies. The use of an aqueous solution of low-cost organic molecules that are stable to oxygen, air, and water implies that a carbon capture system based on this redox cycle has the potential for further development and wider applications.

## Methods

### Procedure for electrochemical reduction of NR in batch

Reactions were carried out with carbon felt (0.5 cm × 0.3 cm × 2 cm was immersed in the solution) cathode and a stainless-steel wire anode in 5 mL H-cell with #9 O-ring equipped with anion exchange membrane. In the cathodic chamber, NRH (58 mg, 0.2 mmol, 50 mM), nicotinamide (488 mg, 4 mmol, 1 M) and lithium perchlorate (212 mg, 2 mmol, 0.5 M) were added into water (4.0 mL). In the anodic chamber equipped with a needle to prevent pressurization was placed lithium perchlorate solution (212 mg, 2 mmol, 0.5 M, 4.0 mL of water). The solution was bubbled with nitrogen for 10 min, after which the electrochemical potential was applied at room temperature by a constant current of −50 mA for 696 s (0.36 mmol of electrons).

### Procedure for electrochemical oxidation of NRH2 solution in batch

Reactions were carried out with carbon felt (0.5 cm × 0.3 cm × 2 cm was immersed in the solution) anode and a stainless-steel wire cathode in 5 mL H-cell with #9 O-ring equipped with anion exchange membrane. In the anodic chamber, the NRH2 solution (50 mM, 4 mL) was bubbled by 15% $CO_2$ for 12 min. In the cathodic chamber equipped with a needle to prevent pressurization was placed lithium perchlorate solution (212 mg, 2 mmol, 0.5 M, 4.0 mL of water). The electrochemical potential was applied at room temperature by a constant current of 50 mA for 696 s (0.36 mmol of electrons). The gas output from the anodic chamber was measured by a flow meter and FT-IR $CO_2$ sensor.

### Procedure for $CO_2$ absorption experiments of NRH2 solution

Procedure for electrochemical reduction of NR in batch was followed with an electrochemical potential of constant current of −50 mA for 773 s (0.40 mmol of electrons) to provide 50 mM NRH2 solution. A 1 mL of the solution was transferred to a 10 mL vial that was prefilled with 1, 4, and 15% $CO_2$ (balanced by nitrogen) and bubbled at a flow rate of 3.33 mL/min. An additional experiment with 15% $CO_2$ at a lower flow rate of 1 mL/min was carried out. The gas output from the vial was measured by a flow meter and FT-IR $CO_2$ sensor.

### Procedure for $CO_2$ absorption experiments of EDA and MEA solutions

Aqueous solutions of EDA (50 mM in 1 mL of water) and MEA (50 mM in 1 mL of water) were transferred to a 10 mL vial that was prefilled with 1, 4, and 15% $CO_2$ and bubbled at a flow rate of 3.33 mL/min. The gas output from the vial was measured by a flow meter and FT-IR $CO_2$ sensor.

### Procedure for $CO_2$ absorption and release experiments using ambient air in batch

The procedure for electrochemical reduction of NR in the batch was followed to provide NRH2 solution (50 mM, 4 mL). The solution was transferred to a 20 mL vial and bubbled with air from the in-house supply at ca. 120 mL/min. After contacting to air, the solution was transferred to the anodic chamber and the procedure for electrochemical oxidation was followed. The gas output from the anodic chamber was measured by a flow meter and FT-IR $CO_2$ sensor.

### Procedure for pH measurement during $CO_2$ absorption

Procedure for electrochemical reduction of NR in the batch was followed to provide NRH2 solution (50 mM, 5 mL). The 5 mL of solution was transferred to an 8 mL vial equipped with a pH probe and bubbled with 1, 4, and 15% $CO_2$ (balanced by nitrogen) at a flow rate of 100 mL/min and air at a flow rate of ca. 120 mL/min.

### Procedure for stability tests using UV−vis absorption spectroscopy

Procedure for electrochemical reduction of NR in the batch was followed with an electrochemical potential of constant current of at −50 mA for 773 s (0.40 mmol of electrons) to provide 50 mM NRH2 solution. A set of 1 mL of solution in an 8 mL vial was prepared and contacted with oxygen (flow rate of 3 mL/min), nitrogen (flow rate of 3 mL/min), $CO_2$ (flow rate of 3 mL/min), and air (flow rate of 5 mL/min). The samples were measured by UV−vis after 30 min, 2 h, and 1 d. Each time 10 μL of the samples were collected and diluted with 10 mL phosphate buffer solution (pH 6.6) to provide 50 μM solutions. After 1 day, the solutions were sealed under the contracted gas and stored at room temperature for 1 week. The 10 μL samples were collected and diluted with 10 mL phosphate buffer solution (pH 6.6) to provide 50 μM solutions that were measured by UV−vis absorption spectroscopy.

### Procedure for electrochemical capture and release of $CO_2$ in a continuous flow using 15% $CO_2$

Reactions were carried out with carbon felt (1.5 cm × 0.3 cm × 1.5 cm was immersed in the solution) cathode and a stainless-steel wire anode in 20 mL H-cell with #20 O-ring equipped with anion exchange membrane. In the cathodic chamber, NRH (289 mg, 1 mmol, 50 mM), nicotinamide (2.4 g, 20 mmol, 1 M), and potassium chloride (1.5 g, 20 mmol, 1 M) were added into water (20 mL). In the anodic chamber equipped with a needle to prevent pressurization was placed potassium chloride solution (1.5 g, 20 mmol, 1 M, 20 mL water). The electrochemical potential was applied at room temperature by a constant current of −100 mA for 1739 s (1.8 mmol of electrons) to provide 20 mL of 45 mM NRH2 solution.

The 15 mL NR solution (50 mM, 1 M NA, 1 M KCl) was added to a 50 mL round bottom flask anolyte reservoir. The 15 mL NRH2 solution was transferred to a 100 mL round bottom flask catholyte reservoir equipped with a stir bar. The solution was bubbled by 15% $CO_2$ (balanced by nitrogen) at a flow rate of 70 mL/min for 15 min.

The peristaltic pump equipped with Masterflex® 14 tubing was set to 1.6 rpm for the liquid flow rate of 0.349 mL/min providing 6.3 min of residence time in each 2.2 mL chamber of the flow cell. After one residence time passed from pumping the solution, the electrochemical potential at a constant current mode of 50 mA was applied to the cell. During operation, a flow rate of 15% $CO_2$ gas stream was maintained at 20 mL/min. The output gas flow from the anolyte reservoir was measured by a flow meter and $CO_2$ sensor.

### Procedure for electrochemical capture and release of $CO_2$ in continuous flow using air

The 30 mL NR solution (50 mM, 1 M NA, 1 M KCl) was added to a 50 mL round bottom flask anolyte reservoir. The 30 mL of NRH2 solution was

transferred to a 500 mL three-neck round bottom flask catholyte reservoir equipped with a stir bar. The solution was bubbled by ambient air at a flow rate of ca. 1000 mL/min for 12 h to saturate the solution. The solution pH was monitored by in situ pH probe simultaneously. The pH was maintained between 9 and 10 during the experiment.

The peristaltic pump equipped with Masterflex® 14 tubing was set to 1.0 rpm for the liquid flow rate of 0.218 mL/min providing 10.1 min of residence time in each 2.2 mL chamber of the flow cell. After one residence time passed from pumping the solution, the electrochemical potential at a constant current mode of 30 mA was applied to the cell. During operation, a flow rate of air was maintained at ca. 1000 mL/min. The output gas flow from the anolyte reservoir was measured by a flow meter and $CO_2$ sensor. A syringe pump was used to add water into the catholyte reservoir at a flow rate of 1.5 mL/hour.

## Data availability

The authors declare that the main data supporting the findings of this study, including experimental procedures and compound characterization, are available within the article and its Supplementary information files, and also are available from the corresponding authors.

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

## Acknowledgements

We thank Dr. Bing Yan for the discussion on the electron transfer mechanism. This work was partially supported jointly by the U.S. Department of Energy, Office of Science, Office of Basic Energy Sciences, Divisions of Chemical Sciences, Geosciences, and Biosciences (CSGB), and Materials Sciences and Engineering (MSE) under FWP 76830.

## Author contributions

H.S. conceived the idea, designed the research, and performed the experiments. H.S. and T.A.H. wrote the manuscript, commented on the final draft of the manuscript, and contributed to the analysis and interpretation of the data.

## Competing interests

H.S. and T.A.H. are inventors on a patent application related to this work filed by the Massachusetts Institute of Technology (No.: 63/355,701; filed on 27 June 2022).
