## [Peer Review File · Nature Communications]

Electrochemical direct air capture of CO₂ using neutral red as reversible redox active materialREVIEWER COMMENTS

Reviewer #1 (Remarks to the Author):

In this work, Seo and Hatton describe the use of the dye neutral red as an efficient mediator for direct air CO₂ capture. According to the authors, the use of this class of redox-active material can improve remarkably the capture cycle, due to their insensitivity to oxygen and high durability. The paper is well written and complete, with informative Electronic Supporting Information.

I recommend publication in Nature Communication as soon as the authors address the following comments:

- The authors employed nicotinamide (NA) to improve the solubility of NR. Have the authors conducted blank experiments with the additive only to prove its innocence in the system? This is an important set of data that is missing, and it is essential to rule out this possibility to exploit the redox-active material at their best capacity
- In the supporting information, the authors screened other different redox-active materials. I think this information should be included in Figure 1 in the main text, as it would help the reader to understand better how NR has been selected for the process.
- It is not clear to the reader why different supportive electrolytes have been employed (LiClO₄ and KCl).
- The authors should report the pictures of the divided-cell batch setup in the supporting information.
- Even if it may seem trivial the difference between the simple amines tested and NR, the authors should explain why the employment of EDA and MEA is not recommended for these processes.
- I think the authors should comment on the scale-up feasibility of this process. CO₂ capture is a very important topic nowadays, and the development of this kind of systems on scale may be an answer to the pressing demand for CO₂ emission control.

Reviewer #2 (Remarks to the Author):

This manuscript describes the setup of a continuous flow electrochemical cell for direct air capture (DAC). Electrochemically basified leuco-neutral red (NRH₂) solution was used as the catholyte for CO₂ absorption, which was more resistant to O₂ compared to other organic redox-active compounds such as phenothiazines. With air as the feed, the NRH₂ solution exhibited a slower absorption kinetics than ethylene diamine but a similar absorption capacity. More importantly, a minimum energy of 64 kJ/mol of CO₂ was demonstrated in the flow cell. In general, the topic is of interest for Nature Communications, but I cannot recommend the publication of this manuscript in its current form.

Review comments include the following:

1. Albeit the improved resistance to O₂, the stability of the electrochemical flow cell is still of concern. Figure 6e shows a reducing electron utilization after ca. 38 h, which does not warrant the long-term operation. Do the authors have data from extended stability tests?
2. In connection with the previous point, did the authors characterize the NRH₂ solution after the stability tests? On Page S11 in the SI, ¹H NMR spectra were presented for the catholyte and anolyte after the 45-h operation. However, the NMR spectra were neither compared with the pristine electrolytes nor discussed in the manuscript.
3. On Page 13, Lines 221–222, the authors mentioned that precipitation of NRH₂. The NMR evidence should be added in the SI. In addition, was it one of the causes for the reduced electron utilization?

In addition, a careful language editing is recommended to improve the writing quality. Listed below are several improper expressions and typographical errors:

1. Page 3, Line 64: "...in the redox process due to the system operates at a pH range of 6–12", the use of "due to" was incorrect.
2. Page 4, Line 83: "Although minor differences between the CV curve shapes we obtained in Figure 1c, we...", the "we" should be "were".

3. Figure 2c: Marks for spelling check were still present. The title of horizontal axis should be placed on the bottom.
4. Page 13, Line 226: It should be "Figure 5e".

Response to Reviewers

Electrochemical direct air capture of CO₂ using neutral red as reversible redox active material

Hyowon Seo and T. Alan Hatton
Department of Chemical Engineering
Massachusetts Institute of Technology
Cambridge MA 02139

Reviewer 1

Comments:

In this work, Seo and Hatton describe the use of the dye neutral red as an efficient mediator for direct air CO₂ capture. According to the authors, the use of this class of redox-active material can improve remarkably the capture cycle, due to their insensitivity to oxygen and high durability. The paper is well written and complete, with informative Electronic Supporting Information. I recommend publication in Nature Communication as soon as the authors address the following comments:

- The authors employed nicotinamide (NA) to improve the solubility of NR. Have the authors conducted blank experiments with the additive only to prove its innocence in the system? This is an important set of data that is missing, and it is essential to rule out this possibility to exploit the redox-active material at their best capacity

Thank you for raising this important point. We have added the blank experiments on page S12 in the SI (page S17). The solution containing 1 M NA and 0.5 M LiClO₄ as a supporting electrolyte was electrochemically reduced at a constant current of -50 mA. The resulting solution was analyzed by ¹H-NMR; the spectrum shows the formation of small amounts of a reductive dimer. As this NA dimer was not detected in experiments using NR (Figure S2), we were able to rule out that the possibility of NA participation in the redox processes under the current conditions.

Figure S12. ¹H-NMR spectra of the solution after reduction of the solution containing 1 M NA and 0.5 M LiClO₄.

In addition, we measured the applied potential vs. Ag/AgCl of the solution with NR and without NR (blank solution) at a various current (2 mA, 5 mA, and 10 mA) in an H-cell. This DC method can represent actual potentials required including not only thermodynamic minimum required potential, but also electron transfer overpotential, surface overpotential, current density overpotential, and solution overpotential of the half cell. As we compared in Figure S13 in the SI (page S18), the applied potentials were 680-900 mV larger in the solution containing only 1 M NA and LiClO₄ than the solution containing NR. Due to the high potential difference to electrochemically reduce NR and NA in an aqueous electrolyte solution, we believe that NR was preferentially reduced in the constant current mode of operation.

Figure S13. Applied potentials vs Ag/AgCl of the solution containing 50 mM NR, 1 M NA, and 1 M KCl in water (blue squares), and 1 M NA and 1 M KCl in water (red dots) at constant currents of 2 mA, 5 mA, and 10 mA. The potential was recorded after 180 seconds of operation. Carbon felt (0.5 cm X 0.3 cm X 2 cm immersed in the solution) was used as a working electrode.

- In the supporting information, the authors screened other different redox-active materials. I think this information should be included in Figure 1 in the main text, as it would help the reader to understand better how NR has been selected for the process.

We thank the reviewer for their thorough review and appreciate the suggestions that have improved the manuscript. We have added Figure 1a with text in caption accordingly.

Figure 1a summarizes air-insensitivity of NR relative to that of several commercially available phenazine and phenothiazine dye compounds. As displayed in the UV-vis spectra, the NRH₂ peak at 455 nm remained constant after 2 hours of air bubbling, which suggested the air-insensitivity of this compound. On the other hand, we did observe rapid reoxidation of the reduced toluidine blue and thionin within 10 min when contacted with air, as their UV-vis peaks of the oxidized forms were recovered.

In caption for Figure 1:

“... (a) UV-vis spectra for NR, toluidine blue (TB), and thionin (TN) in an aqueous solution during tests for air-sensitivity. Solutions containing the reduced organic dye compounds (50 mM, 1 mL) were bubbled with air for 10 min (TBH, TNH) or 2 hours (NRH2) at a flow rate of 3 mL/min. ...”

Line 54:

“... Here, we report electrochemical direct air capture of CO₂ using neutral red (NR), a commercial organic dye molecule, as an oxygen insensitive organic redox-active compound (Figure 1a) in the presence of nicotinamide (NA) as a hydrotropic agent to increase its solubility in the aqueous system (Figure 1b).²⁵ Other commercial phenothiazine compounds such as toluidine blue (TB), and thionin (TN) could not satisfy the need for oxygen-insensitivity (Figure 1a), and therefore were not considered further for this task.”

- It is not clear to the reader why different supportive electrolytes have been employed (LiClO₄ and KCl).

We used lithium perchlorate (LiClO₄) as our choice of supporting electrolyte for cyclic voltammetry because it has a wide electrochemical potential window, chemical stability, pH insensitivity, good solubility, and good conductivity in water. Conventionally, LiClO₄ is one of the preferred choices for electroanalytical experiments for the above reasons.

Although LiClO₄ was a good choice for the electroanalytical experiments and the small batch reaction, we changed to potassium chloride because of its better conductivity (28.3 mS/cm in 1 M LiClO₄ and 1 M NA in water vs 43.7 mS in 1 M KCl and 1 M NA in water) and higher solubility of NR in water (50 mM in 0.5 M LiClO₄ and 1 M NA vs 306 mM in 0.5 M KCl and 1 M NA) to avoid potential clogging of the flow system.

We have added conductivity and solubility information in line 266 in the manuscript.

“We constructed a continuous flow cell to process 50 mM NR solution in the presence of 1 M NA and 1 M KCl as a supporting electrolyte due to its better conductivity than that of LiClO_4 (28.3 mS/cm in 1 M LiClO_4 and 1 M NA in water vs 43.7 mS in 1 M KCl and 1 M NA in water) and higher solubility of neutral red in water (50 mM in 0.5 M LiClO_4 and 1 M NA vs 306 mM in 0.5 M KCl and 1 M NA) to avoid potential clogging of the tubing in the flow system.”

- The authors should report the pictures of the divided-cell batch setup in the supporting information.

We have added the pictures of the divided H-cell batch setup in Figure S5 in the SI (page S12).

Figure S5. Picture of H-cell

- Even if it may seem trivial the difference between the simple amines tested and NR, the authors should explain why the employment of EDA and MEA is not recommended for these processes.

We thank the reviewer for this suggestion. We have added a sentence to explain why the employment of EDA and MEA is not recommended in this process.

In line 178:

“... We also compared the NR CO₂ absorption profiles with those of 50 mM solutions of amines ethylenediamine (EDA) and monoethanolamine (MEA), which are traditionally used for CO₂ capture (Figures 4d to f); these electrochemically inactive amines are frequently used in a comparative studies of CO₂ absorption.^{27,28} ...”

• I think the authors should comment on the scale-up feasibility of this process. CO₂ capture is a very important topic nowadays, and the development of this kind of systems on scale may be an answer to the pressing demand for CO₂ emission control.

We thank the reviewer for their comments and suggestions. We have added a paragraph about the scale-up feasibility of this process in the manuscript.

In line 307:

“Although further optimization and engineering efforts are required for scale-up, the NR/NRH₂ redox couple meets several essential criteria for implementation in carbon capture processes, including redox stability, reasonable solubility in water, fast kinetics, and oxygen insensitivity. In addition, to operate this system with the minimum potential required on an industrial scale, it will be necessary to increase the current density and reduce the overpotential by optimizing the design of the cell structure and material.”

Reviewer #2

Comments:

This manuscript describes the setup of a continuous flow electrochemical cell for direct air capture (DAC). Electrochemically basified leuco-neutral red (NRH₂) solution was used as the catholyte for CO₂ absorption, which was more resistant to O₂ compared to other organic redox-active compounds such as phenothiazines. With air as the feed, the NRH₂ solution exhibited a slower absorption kinetics than ethylene diamine but a similar absorption capacity. More importantly, a minimum energy of 64 kJ/mol of CO₂ was demonstrated in the flow cell. In general, the topic is of interest for Nature Communications, but I cannot recommend the publication of this manuscript in its current form. Review comments include the following:

1. Albeit the improved resistance to O₂, the stability of the electrochemical flow cell is still of concern. Figure 6e shows a reducing electron utilization after ca. 38 h, which does not warrant the long-term operation. Do the authors have data from extended stability tests?

We agree with the reviewer that long-term stability is an issue, but note that our results are on a par with the best results reported to date. The apparent loss of electron efficiency in our experiments can be put down to clogging of the needle used for introduction of the air to the cathode chamber. We have added additional results of a longer time operation for 96 h and replaced Figure 6d and e with the updated ones. The time/circulation time in this continuous flow system was over 20 without significant loss of electron utilization. Compared to similar recent publications (references 9 and 10 in the manuscript), this number of redox cycles is promising value to continue studying for the large-scale system with longer operation hours.

The reduction in electron utilization after ca. 38 h in Figure 6e in the manuscript was due to partial clogging of the needle used to introduce air to the catholyte solution; the high air flow rate (ca. 1000 mL/min) quickly dried water at the tip of the needle, leading to precipitation of solution components, most likely NR, NA, and KCl. Indeed, we repeatedly observed a reduction in electron utilization every 24-48 hours due to this air needle clogging. On replacement of the air needle, we observed that the electron utilization increased again to the range of 0.3-0.4. The yellow triangle marks in Figure 6e indicate when the air needle was replaced. We have added this explanation in the manuscript.

In line 297:

“...The reduction in electron utilization after ca. 43 h was due to partial clogging of the needle used to introduce air to the catholyte solution; the high air flow rate (ca. 1000 mL/min) quickly dried water at the tip of the needle, leading to precipitation of solution components, most likely NR, NA, and KCl. Indeed, we repeatedly observed a reduction in electron utilization every 24-48 hours due to this air needle clogging. On replacement of the air needle, we observed that the electron utilization increased again to the range of 0.3-0.4....”

Figure 6. ... (d) Released CO₂ amount over time using ambient air as the feed gas. (e) Electron utilization over time with an ambient air feed. Electrolytes comprised 30 mL of 50 mM NRH₂ in 1 M NA and 1 M KCl and 30 mL of 50 mM NR in 1 M NA and 1 M KCl in water. The red curve indicates the average value of electron utilization for each one-way travel (circulation time/2). The numbers in red indicate the time/circulation time. The yellow triangle marks indicate where the air needle was replaced.

In addition to potential needle clogging, we were able to ascertain that the absorption unit can be improved for the system stability in the future. Due to the slow absorption of CO₂ from the air (410 ppm), high flow rate of air was necessary to saturate the catholyte solution (pH maintained between 9 to 10). In our lab scale demonstration, we used ca. 1000 mL/min of air flow for the operation at 30 mA current and 20-30 mL of catholyte solution. Due to the rapid bubbling of the solution, splashing of the catholyte and evaporation of water may have caused degradation in the lifetime of the cell. We believe that better CO₂ absorption unit design can improve CO₂ absorption rate as well as the electrochemical cell's operation time in the future. We have added this point in the manuscript.

In line 333:

“... We were able to ascertain that the absorption unit can be improved for the system stability in the future. Due to the slow absorption of CO₂ from the air (410 ppm), high flow rate of air was necessary to saturate the catholyte solution. In our lab scale demonstration, we used ca. 1000

mL/min of air flow for the operation at 30 mA current and 20-30 mL of catholyte solution. Due to the rapid bubbling of the solution, splashing of the catholyte and evaporation of water may have caused degradation in the lifetime of the cell. We believe that better CO₂ absorption unit design can improve CO₂ absorption rate as well as the electrochemical cell's operation time in the future.....”

2. In connection with the previous point, did the authors characterize the NRH2 solution after the stability tests? On Page S11 in the SI, ¹H-NMR spectra were presented for the catholyte and anolyte after the 45-h operation. However, the NMR spectra were neither compared with the pristine electrolytes nor discussed in the manuscript.

We thank the reviewer for this astute observation. We have characterized the NRH2 solution by UV-vis, which shows the characteristic absorption peak of NRH2 at 455 nm and peaks for NR at 460 and 550 nm after the stability tests as in Figure 5.

In addition, we have filtered the solutions after a week under N₂, O₂, CO₂, and air, and characterized the filtered solid using ¹H-NMR to show the stability of NRH2 under stability test conditions (Figure S23, page S27, SI). The precipitated solids are mainly NRH2 and NA without observation of reoxidized NR. This result is consistent with our UV-vis analysis of the filtrate in Figure 5.

Figure S23. ¹H-NMR spectrum of precipitate after stability tests under O₂, N₂, CO₂, and air for a week. Minor peaks from byproduct were observed at δ 8.94, 8.44, 8.09, 7.28 under O₂ and N₂.

We have updated the $^1\text{H-NMR}$ spectra after continuous flow operation using air for 96 h with the pristine electrolytes in Figure S4 (page S11, SI) and included discussion of this in the manuscript as below.

In line 304:

“...After operation of direct air capture under continuous flow, the anolyte and catholyte solutions were analyzed by $^1\text{H-NMR}$ to confirm no participation of NA in this redox process (Supplementary Fig. S4)...”

Figure S4. ^1H NMR spectra were collected after 96 hours of operation under air confirming redox resistance of nicotinamide under the current conditions.

3. On Page 13, Lines 221–222, the authors mentioned that precipitation of NRH₂. The NMR evidence should be added in the SI. In addition, was it one of the causes for the reduced electron utilization?

We have added the $^1\text{H-NMR}$ spectra of the precipitated NRH₂ in Figure S23 in the SI (page S27). In addition to peaks from NRH₂ and NA, we were able to observe that minor peaks at δ 8.94, 8.44, 8.09, 7.28 under O₂ and N₂, which might be derived from NA. We are currently working on

the identification of potential byproduct formation pathways to improve the cell's stability and operation hours.

Although further studies are warranted, we currently believe that the precipitation is one of the causes for the reduced electron utilization, albeit a minor one. We have added this point in the manuscript.

In line 232:

“...due to its low solubility under the current condition led UV-vis peak degradation (Supplementary Figure S23). In addition to peaks from NRH₂ and NA, minor peaks potentially derived from NA were observed under O₂ and N₂. Although further studies are warranted on the byproduct formation pathways, we currently believe that the precipitation is one of the causes for the reduced electron utilization, albeit a minor one....”

In addition, a careful language editing is recommended to improve the writing quality. Listed below are several improper expressions and typographical errors:

1. Page 3, Line 64: “...in the redox process due to the system operates at a pH range of 6–12”, the use of “due to” was incorrect.

We have changed the sentence for clarity as below.

“...The conjugate base NR would mainly participate in the redox process because the system operates in the pH range of 6-12...”

2. Page 4, Line 83: “Although minor differences between the CV curve shapes we obtained in Figure 1c, we...”, the “we” should be “were”.

We have changed the sentence as below.

“...CV curves of NR shows quasi-reversibility in the redox-activity of NR/NRH₂ in the presence and absence of NA. ...”

3. Figure 2c: Marks for spelling check were still present. The title of horizontal axis should be placed on the bottom.

We have updated the figure 2c with horizontal axis on the bottom. We also have removed the spelling check marks.

4. Page 13, Line 226: It should be “Figure 5e”.

We have changed the sentence as below.

“...A third set of UV–vis absorption measurements (Figure 5e), this time on solutions contacted with CO₂, was carried out...”

REVIEWERS' COMMENTS

Reviewer #1 (Remarks to the Author):

I am highly satisfied with the work conducted by the authors to implement the suggestions and clarify important points of the manuscript.

I would recommend the publication of the paper in its current form.

Reviewer #2 (Remarks to the Author):

The revision is satisfactory, and I recommend it to be published on Nature Communications.